# Development and Evaluation of an Antiviral Agent Medication Adherence Education Program for Patients with Chronic Hepatitis C

**DOI:** 10.3390/ijerph17186518

**Published:** 2020-09-07

**Authors:** Hoo Jeung Cho, Euna Park

**Affiliations:** Department of Nursing, Pukyong National University, Busan 48513, Korea; ccu0401@naver.com

**Keywords:** medication adherence, hepatitis C, antiviral agents, nursing, education

## Abstract

This study aimed to develop and evaluate a goal attainment theory-based antiviral agent medication adherence education program (AMAEP) for patients with chronic hepatitis C. A nonequivalent control group pretest-posttest design was used. Data were collected from December 2019 to March 2020 from a control group of 35 outpatients and an experimental group of 28 outpatients older than 20 years old who had been diagnosed with chronic hepatitis C. The data analysis included an independent *t*-test, a χ^2^-test or Fisher’s exact test, a Kolmogorov–Smirnov test, an analysis of covariance, and a Mann–Whitney U test. The results showed the effectiveness of the education program for patients with chronic hepatitis C. There were significant differences between the control group and experimental group in patients’ knowledge of chronic hepatitis C (Z = −5.91, *p* < 0.001), medication self-efficacy (Z = −5.02, *p* < 0.001), medication adherence rate (*t* = −3.88, *p* < 0.001), medication misuse behavior (Z = −5.00, *p* < 0.001), and patients’ satisfaction with their interaction with healthcare practitioners (Z = −6.61, *p* < 0.001). Therefore, we hope that the education program developed in this study will be utilized as an intervention for patients with chronic hepatitis C and be further developed for other patients with viral hepatitis.

## 1. Introduction

In 2018, hepatitis C was classified as a group 3 legal communicable disease in Korea [1], and the number of reported cases increased rapidly from 6396 reported cases in 2017 to 10,811 cases in 2018 (an increase of about 69%) [2]. About 50–80% of patients infected with the hepatitis C virus (HCV) progress to a chronic infection stage, which can cause chronic liver damage and lead to cirrhosis of the liver and liver cancer [3]. This highlights the importance of managing the disease appropriately.

Direct acting antivirals (DAA) are oral antiviral agents used to treat chronic hepatitis C (CHC). The goal of this treatment is to reach a state of sustained virologic response for at least 12 weeks after completion of the treatment and for the patient’s blood to then be free of HCV ribonucleic acid (RNA) [3]. HCV treatment also leads to a significant reduction in liver-related morbidity and mortality [4]; therefore, an appropriate treatment must be recommended to ensure that patients reach their treatment goals.

It has previously been reported that oral DAA may cause side effects when combined with ribavirin [5]; moreover, in chronic diseases, patients often become fatigued from long-term medication use or lose the motivation to take their medication [6]. Self-efficacy is the main variable in medicine administration and patients with higher self-efficacy tend to be more efficient in adhering to their treatment, which affects the treatment results [7]. Since not taking medication leads to discontinuation of the treatment, it is important to actively monitor the treatment process, particularly medication use among patients [8,9]. Misuse has been defined as the incorrect use of an over-the-counter (OTC) product for a medical purpose, usually in regard to dosage or duration of use [10]. Although OTC preparations may be relatively safe for healthy younger adults, they often create problems for chronic disease patients, particularly with pathologic conditions and combined with other substances (e.g., nonsteroidal anti-inflammatory drugs, herbs) [10]. Therefore, it is crucial for patients to communicate with a medical professional before taking OTC medications [11] and for patients to be educated about their disease because patients with more knowledge of their disease and better awareness of the state of their health tend to be more actively engaged in their treatment [12].

The objective of this study was to promote the implementation of antiviral treatment through the interaction between the patient and the healthcare practitioner based on King’s theory of goal attainment [13]. According to this theory, interaction and goal attainment are induced by communication between the healthcare practitioner and the participant (or patient). Education by the healthcare professional and learning by the participant involves self-activities that are achieved through sensory perception, conceptualization, and critical thinking [14]. The relationship between the patient and the healthcare practitioner is a fundamental factor in managing chronic diseases and improving the quality of this relationship is essential for successful treatment [15]. To increase mutual satisfaction, the healthcare practitioner should respect their patients and provide knowledge concerning their disease that is appropriate to their level of understanding [16,17].

Considering the above, this study developed an AMAEP based on goal attainment theory for patients with CHC who take oral antiviral drugs and then compared the results from an experimental group and a control group before and after the intervention to evaluate the effectiveness of the program.

## 2. Materials and Methods

### 2.1. Study Design and Participants

This study follows a nonequivalent control group pretest-post-test design. First, an AMAEP based on the theory of goal attainment was developed. Then, the AMAEP was verified by applying it to patients with CHC and confirming its efficacy through an assessment of patients’ knowledge of CHC, self-efficacy in taking antiviral drugs, medication adherence rate, medication misuse behavior, and patients’ satisfaction with their interaction with healthcare practitioners. To prevent treatment spillover effects, patients in general hospital A were selected for the control group and patients in general hospital B for the experimental group. We included patients with CHC that received oral antiviral treatment in the outpatient department of gastroenterology at hospitals A and B located in a metropolitan city from 19 December 2019 to 30 March 2020. The selection criteria were as follows: (a) adults aged 20 years or older, (b) patients with serological examination result of HCV antibody (+) or HCV polymerase chain reaction (PCR) qualitative analysis (+), (c) patients who have been diagnosed with hepatitis C by a gastroenterologist more than six months ago, (d) patients receiving oral antiviral drug combination therapy, (e) patients who can read and understand Korean and have no communication difficulties, and (f) patients with a Class A Child-Turcotte-Pugh (CTP) score. The CTP score is a widely used criteria to determine the severity of liver injury or presence of fibrosis. Patients infected with both HIV and HCV were excluded. The number of participants in this study was calculated using G*power 3.1 software (Heinrich Heine University Düsseldorf, Düsseldorf, Germany) [18]. The two-tailed *t*-test showed a significance level of 0.05, a power (1-β) of 0.80, and an effect size d = 0.80. These results indicated a minimum participant group size of 26 patients per group.

For the experimental group, 33 patients with CHC treated at hospital B from December 2019 to January 2020 met the selection criteria and participated in the pre-test. However, three patients withdrew after attending three of the six sessions due to personal circumstances (disease) and two patients withdrew consent, leading to a 15.2% dropout rate. The total number of participants in the experimental group was 28 (Figure 1).

For the control group, 35 patients with CHC treated at hospital A from December 2019 to January 2020 met the selection criteria and participated in the pre-test. There were no dropouts and all the participants participated for the entire duration of the program including the post-test (Figure 1).

### 2.2. Antiviral Agent Medication Adherence Education Program (AMAEP)

The AMAEP in this study was developed based on King’s goal attainment theory.

Starting in January 2019, the existing literature was reviewed and pre-tests (interviews with participants) were analyzed. The program was developed to include various subjects such as knowledge regarding CHC, antiviral medication adherence, precautions when taking antiviral drugs and interaction between the drugs, and health management after the treatment.

The main protocol of the program centered around the concept of health coaching, an intervention method that helps patients to acquire knowledge, skills, and coping strategies to manage the disease on a self-directed basis. By improving patients’ capabilities in a mutual relationship with the coach, the protocol emphasizes the patients’ autonomy by considering their choices and responsibilities [19]. Health coaching and its ability to instill specific behavioral changes through planning and achieving health-related goals are consistent with King’s (1981) theory of goal attainment [14]. To voluntarily induce individual behavioral change through health coaching, group coaching [20], and one-on-one coaching, especially via telephone [21], has been shown to be helpful; therefore, this study applied one-on-one education and phone counseling as intervention methods while considering the needs and accessibility of the participants.

A booklet containing information about the AMAEP and a drug journal to enhance patients’ medication adherence were developed. The booklet comprised information on hepatitis C, on the antiviral treatment that patients would receive, precautions related to taking the drugs, and information on health management. The booklet was prepared with reference to previous literature [1,22] and was amended and supplemented based on advice from seven healthcare experts: two nursing professors, one gastroenterologist, three nurses from the department of gastroenterology, and one pharmacist. Based on feedback regarding the education time and method, the program contents were reviewed and modified.

As this program is aimed at patients with CHC who are taking antiviral drugs, it was tailored according to patients’ outpatient visits and education sessions were conducted in a quiet space that allowed for individual conversations. Based on an existing self-management program for patients with CHC [23] that was effective within six sessions, this program was likewise developed to consist of six sessions. The outpatient visits included three face-to-face sessions (two individual sessions and one group session) and three telephone sessions during which the effects and side effects of the drugs were evaluated and the hepatitis C virus tests and serological tests were conducted and monitored [24].

Individual sessions consisted of a program of face-to-face sessions, phone calls, text messages, and social network communications for a total of six sessions during eight weeks. Group sessions consisted of four discussion sessions of 2–5 participants per group, according to the schedule of the participants that consented. In the fifth session, a social network chatroom was created to share questions and answers and participants’ experiences regarding antiviral drug treatment and disease management (Table 1).

### 2.3. Study Protocol and Intervention

The program for the experimental group was conducted in private by a researcher in the outpatient conference room from 6 January 2020 to 30 March 2020. The researcher introduced the AMAEP and focused on individual interactions with the patients.

The assistant researcher was given an hour of training twice per week. The training focused on the purpose of the program, the enrollment of subjects in both the experimental and control group, and the content related to the preliminary and follow-up investigations. To reduce any bias that might affect the results, only the principal researcher participated in the education program.

The study protocol was as follows. A notice of recruitment to participate in the education program for patients with CHC was posted on the bulletin board of the Department of Gastroenterology in hospitals A and B. Patients interested in participating in the program after outpatient visits were convenience sampled. Participants who met the inclusion and exclusion criteria consented to participate in the program. The participants agreed to the individual sessions, in which a nurse interacted with them one-on-one and assessed any problems related to disease treatment and management. Participants then worked to achieve the program’s goals, which included improving disease-related knowledge, improving self-efficacy in taking antiviral drugs, improving the rate of antiviral drug use, reducing drug misuse, and improving satisfaction with interactions with the clinicians. In the individual sessions, the nurse provided education on informational aspects (hepatitis C, antiviral drugs, precautions related to taking antiviral drugs, and health management), and encouraged continued participation and adherence to the antiviral medication.

In the first, fourth, and sixth face-to-face sessions, drug information was compared to increase antiviral medication adherence, and to determine whether the medication schedule was well adhered to, as confirmed by the drug journal and pill counts. In addition, questions were asked to confirm participants’ acquisition and understanding of the information provided at each session in order to achieve proper disease and health management. The second, third, and fifth sessions were telephone conversations involving a simple exchange of information related to the topic and confirmation of whether the medication had been taken.

It is important that the needs of individuals in mutual relationships are satisfied in order to ensure smooth individual and group interactions. During the program, the researcher provided immediate answers to the participants’ inquiries or requests by telephone or text messages and also later re-confirmed that the participants’ problem was solved. These one-on-one interactions were valuable for identifying individual needs and for personalized interventions. Group interactions helped to support the implementation of the treatment and health management by allowing the participants to share their experiences regarding the disease and to form relationships.

The control group received the usual care at the hospital during the intervention period. After the experimental group completed the 8-week AMAEP, the control group received the same AMAEP for 1 week.

### 2.4. Study Tools

This study used a tool developed by Gupta et al. [25] to measure knowledge of CHC, which Sun and Ju [26] translated into Korean with the approval of the original authors. Gupta et al. [25] did not present a reliability coefficient, and Sun and Ju [26] reported a Cronbach’s α of 0.78. The Kuder-Richardson reliability coefficient (KR-20) in this study was 0.82.

The self-efficacy measurement tool for CHC antiviral agents used in this study was a revised and supplemented version of the HIV Medication Taking Self-Efficacy Scale developed by Erlen et al. [27] and approval from the author was obtained before use. The scale was translated by a Korean and English-speaking nursing major and reviewed by a gastroenterologist. It was then compared to the reverse-translated version of an English major. The Cronbach’s α for Erlen et al. [27] was 0.93 and it was 0.96 in this study.

The most common method used to measure participants’ medication intake is to count the number of pills left in the patient’s vial. This is a simple method that is useful to experienced researchers [28]. The number of pills taken during a specified period was divided by the number of pills that should be taken for a certain period and calculated as a percentage. The closer the percentage was to 100, the higher the antiviral medication adherence rate.

A tool to measure medication misuse behavior was developed by Lee and Park [10] and was used in this study after obtaining approval from the authors. It consists of a total of 13 items: “Increase dosage or frequency when symptom appear”, “Decrease dosage or frequency when symptom disappear”, “Missing dosage”, “Stop medication arbitrarily when symptom disappear”, “Stop medication due to side effects without Dr’s advice”, “For saving medication, decrease dosage or frequency”, “Use non-prescription drug with prescription drug”, “Use other peoples’ drugs”, “Use other peoples’ drugs with prescription drug”, “Use other peoples’ drugs and non-prescription drug with prescription drug”, “Use polypharmacy at least five more”, “Take medication with alcohol and tobacco”, and “Take medication one more time than scheduled”. Responses were classified as either “yes” (coded 1) or “no” (coded 0). A high total score signified a high level of medication misuse behavior. In Lee and Park’s study [10] the Cronbach’s α was 0.70 and the KR-20 reliability coefficient in this study was 0.80.

The tool used in this study to measure patients’ satisfaction with their interaction with healthcare practitioners was the Client Satisfaction Tool developed by Bear and Bowers [29] and translated, modified, and supplemented by Choi and Yoo [30]; the authors approved the use of this tool. The Cronbach’s α in the original study was 0.96, for Choi and Yoo [30] it was 0.87, and in this study, it was 0.93.

### 2.5. Statistical Analysis

The data were analyzed using SPSS WIN 25.0 software (IBM Co., Armonk, NY, USA) and a two-tailed *t*-test was conducted at a significance level of 0.05. The general characteristics, disease-related characteristics, and homogeneity of the dependent variables in the experimental and control groups were analyzed with descriptive statistics and a *t*-test, χ^2^-test, or Fisher’s exact test. The normality of the experimental and the control groups concerning the pre-test was verified using a Kolmogorov–Smirnov test. Variables that were normally distributed were tested for homogeneity via an independent *t*-test, while those that were not normally distributed were tested for homogeneity via a Mann–Whitney *U* test. Knowledge of the disease after the intervention, self-efficacy of taking antiviral drugs, drug misuse, and satisfaction with interaction with the clinicians after the intervention program were tested for normality of the pre-post difference in values between the intervention and the control group using a Kolmogorov–Smirnov test. Moreover, because the variables were not normally distributed, a Mann–Whitney U test was used for the analysis. The effects of AMAEP were analyzed using analysis of covariance after controlling for pretest scores.

### 2.6. Ethical Considerations

Data collection for this study was approved by the Institutional Review Board of P University (IRB approval number: PKNUIRB-1041386-201911-HR-54-02) and was conducted from 19 December 2019 to 30 March 2020 by collecting the data directly from participants through participant recruitment. Patients with CHC who voluntarily participated in the study were selectively sampled as the target population. These participants were specifically informed of the purpose and details of this study and provided with enough time to decide on their participation. Participants were free to withdraw their informed consent to participate at any time during the study. The time required to complete the questionnaire was approximately 25–30 min.

All subjects were given a transportation fee of 50,000 won (Korean currency) upon each visit. During the six program visits, the subjects in the experimental group were given coffee and refreshments (amounting to approximately 10,000 won per person) in addition to the transportation fee.

## 3. Results

### 3.1. General and Disease-Related Characteristics of Participants and Homogeneity Test

The general characteristics of the participants were as follows. In the control group, 16 (45.7%) were male and 19 (54.3%) were female, while in the experimental group, 13 (37.1%) were male and 15 (42.9%) were female. The average age of the control group was 62.91 (standard deviation (SD) = ±9.66) years old, and that of the experimental group was 60.89 (±7.67) years old. The average body mass index (BMI) of the control group was 24.02 (±2.28) kg/m^2^, and that of the experimental group was 23.71 (±2.31) kg/m^2^. The most frequent family structure in the control group was “spouse and children” for 17 participants (48.6%) and that of the experimental group was “spouse” for 13 participants (46.4%). In the control group, 18 participants (51.4%) perceived their economic level to be “low” and in the experimental group, 14 participants (50.0%) perceived their economic level to be “medium.” As for the education level, 17 participants (48.6%) in the control group were “elementary school graduates” and 16 participants (57.1%) were “middle/high school graduates.”

With regard to the participants’ smoking status and alcohol consumption, most participants in the control and the experimental groups had never smoked, with 16 participants (45.7%) and 14 participants (50%), respectively, reporting they were nonsmokers. Most participants in the control (22 participants or 62.9%) and the experimental group (18 participants or 64.3%) drank “to appropriate standards.”

With regard to disease-related characteristics, more than half of the participants (27 participants (77.1%) in the control group and 18 participants (64.3%) in the experimental group) had comorbidities. Most participants in both groups took concomitant drugs, with 23 participants (65.7%) in the control and 17 participants (60.7%) in the experimental group. Serological tests showed that the average aspartate aminotransferase (AST) level was 32.29 (±16.39) IU/L in the control group and 41.68 (±22.13) IU/L in the experimental group, and the average alanine aminotransferase (ALT) level was 38.29 (±20.96) IU/L in the control group and 46.89 (±25.58) IU/L in the experimental group. All participants belonged to Class A with Child-Turcotte-Pugh scores of five. In terms of antiviral drugs, 15 participants (42.9%) in the control group took ombitasvir, paritaprevir, ritonavir, and dasabuvir, and 12 participants (60.7%) in the experimental group took sofosbuvir and ribavirin.

Among the general characteristics, only education level was not homogeneous and showed a statistically significant difference (χ^2^ = 7.13, *p* = 0.028), whereas all other general characteristics and all disease-related characteristics were homogeneous with no significant differences between the two groups (Table 2).

### 3.2. Homogeneity Test of Dependent Variables

The results of the homogeneity test of the dependent variables prior to the intervention are shown in Table 2. Before participating in the program, the patients’ knowledge on the CHC was 16.00 (±4.21) points in the control group and 15.79 (±3.59) points in the experimental group (t = 0.21, *p* = 0.831), while the self-efficacy of taking antiviral drugs was 5.83 (±0.95) in the control group and 5.53 (±0.74) in the experimental group (Z = −1.17, *p* = 0.245). The level of medication misuse behavior was 6.63 (±3.44) points in the control group and 7.25 (±3.12) points in the experimental group (Z = −0.89, *p* = 0.377) and the satisfaction level with interaction with the healthcare practitioner was 20.43 (±4.38) in the control group and 19.93 (±3.05) in the experimental group (Z = −0.46, *p* = 0.650), indicating homogeneity with no significant differences (Table 3).

### 3.3. Effectiveness of the AMAEP between Experimental and Control Groups

Analysis of variance with the pretest as a covariate revealed significantly different changes between the groups in their knowledge about CHC (F = 82.90, *p* < 0.001, η^2^ = 0.580; Table 4), self-efficacy in taking the antiviral agent (F = 55.08, *p* < 0.001, η^2^ = 0.479; Table 4), medication misuse behavior (F = 53.40, *p* < 0.001, η^2^ = 0.471; Table 4), and satisfaction level with their interactions with healthcare practitioners (F = 109.90, *p* < 0.001, η^2^ = 0.647; Table 4). When we calculated the antiviral drug medication adherence as a percentage and average after the intervention, there was a significant difference between the two groups with 95.66 (±4.99) in the control group and 99.08 (±1.32) in the experimental group (t = −3.88, *p* < 0.001; Table 4).

## 4. Discussion

To enable patients with chronic diseases to manage their diseases or implement health care, it is essential for them to obtain information related to the disease, and also to be educated by medical professionals. Below, we discuss the educational method, content, and effectiveness of the AMAEP.

In this study, education and interaction were used to improve the self-management capabilities of patients with CHC who are taking antiviral drugs. Education of patients is said to be one of the most important roles of healthcare professionals [8,14]. From a medical professional’s perspective, patient education is generally provided in a persuasive and instructive way, with nurses delivering instructive counseling centered around assessment and problem-solving. However, there is a need to supplement the education that patients receive. Based on King’s [13] theory of goal attainment, this study aimed to establish and achieve these goals through the interaction between the nurse and the participant. Because nursing is based on dynamic interaction and interpersonal relationships that involve individual perception, judgment, action, and reaction [13], the program is based on education led by the patient through mutual interaction rather than subjective nurse-led education. Previous studies [31,32] have demonstrated that setting a goal through a mutual exchange between the nurse and a participant shows effective results, consistent with the findings of this study.

In this study, the AMAEP for patients with CHC led to increased knowledge of CHC in the experimental group compared with the control group. The level of knowledge of CHC in both groups prior to the intervention was lower than that of participants in previous studies [25,26]. This may be due to a general lack of awareness of the disease because patients with CHC are often asymptomatic, even if they are infected with the HCV [33]. In addition, the level of knowledge may be low because the information provided by medical professionals is often fragmented and communicated in a one-way fashion [34]. Medical professionals need to provide patients with adequate guidance and training in order for patients to understand their infection status and to actively participate in their disease management in the future.

The AMAEP for patients with CHC also improved patients’ self-efficacy in taking antiviral drugs compared with the control group, which is similar to the results of previous studies [9,35,36]. This may be because this program recognized the need for antiviral drugs and confirmed the treatment goals through education, which improved patients’ understanding of the disease. At the same time, it provided encouragement to patients to take their antiviral drugs regularly. Smooth communication between the patient and the healthcare practitioner positively affects the patient’s self-efficacy [16], and we confirmed that the interaction and communication between the medical practitioner and the participants in this program influenced the self-efficacy of the participants.

The AMAEP for patients with CHC improved the medication adherence of the experimental group compared to the control group. The average rate of medication adherence for the participants in the experimental group of this study was 99.08%. This rate is higher than that of Namba et al. [8] that included patients using interferon treatment. This difference may be because the researcher in the current study explained the correct method of taking the drugs and the reason for taking them, and ensured that the practice was carried out every day, rather than simply telling the participants how to take the drug. Specifically, participants were asked to keep a drug journal, and during face-to-face sessions, researchers conducted a pill count and compared the results with the contents of the journal. Education, phone consultation [37], and automatic alarms for taking the drug appear to increase adherence to medication [38]. However, as the measurement of adherence to the medication was self-reported in this study, a future study that uses applications or a digital directly observed treatment (DOT) system using a Smart pillbox [39] might confirm the reliability of the program.

This study also showed a significant reduction in medication misuse behavior in the experimental group after the AMAEP, with a pre- and post-average difference of −3.36. This is similar to the −2.81 point reduction in a study by Lee [11] and the −2.00 point difference in a study by Park et al. [40] after conducting an educational program on drug misuse among older adults. The reduction in drug misuse may have been because patients with liver disease were discouraged from taking various dietary supplements or over-the-counter medications without consulting a healthcare practitioner in a number of sessions and through one-on-one education with the nurses.

Lastly, this study showed that patients in the experimental group were more satisfied with their interactions with healthcare practitioners after the intervention. This is consistent with the work of Namba et al. [8], who showed that an intervention in which a medical professional supports the participants leads to an increased level of satisfaction with the medical practitioner. The relationship between a patient and their healthcare practitioner is one of the most basic factors in healthcare, and improving this mutual relationship is important in managing chronic diseases [15]. Having nurses with at least three years of experience in the department who are familiar with the nature of the disease and the side effects of the treatment conduct the educational intervention may be helpful [8].

This study confirmed that the AMAEP of six sessions based on King’s theory of goal attainment is an effective intervention for patients with CHC in terms of their knowledge of the disease, self-efficacy of taking antiviral drugs, adherence to the medication, drug misuse, and satisfaction with their interactions with medical practitioners.

Considering the limitations and future studies, we acknowledge first, that this study was conducted using convenience sampling in a certain region; therefore, there are limitations to the generalizability of the study. Second, it is necessary to assess the effectiveness of the program more accurately by applying a strict double-blind experiment. Third, the final assessment of medication adherence was conducted after 8 weeks even though the duration of therapy for CHC is often 12 weeks. We only focused on patients’ concurrent medication outcomes, and ignored the long-term effect of the AMAEP protocol. Thus, we still need to further our research to examine the program’s effectiveness and to consider any warranted modifications. Fourth, we suggest using a more structured training program in the future that utilizes nurses who are trained to educate patients through the program or for counseling. In clinical practice, it is rare to find trained nurses who can properly apply the AMAEP. Another limiting factor is the time required to conduct education programs. The factors discussed thus far are likely to pose a challenge to the effectiveness of the AMAEP; therefore, the necessary human resources and an appropriate physical environment are required.

## 5. Conclusions

The AMAEP developed and verified in this study utilized an integrated disease self-management approach by including disease management and health care after the completion of the treatment and it included six individual sessions (face-to-face exchange or telephone interaction) with the patient according to their outpatient schedule. This study showed the effectiveness of an AMAEP for CHC patients with regard to patients’ knowledge of the disease, medication self-efficacy, medication adherence rate, medication misuse behavior, and satisfaction with their interaction with healthcare practitioners. The continuous interactions between nurses and participants as well as those among the participants themselves throughout the intervention proved to be an effective strategy; therefore, medical practitioners should devise ways to provide face-to-face counseling and feedback to their patients.

## Figures and Tables

**Figure 1 ijerph-17-06518-f001:**
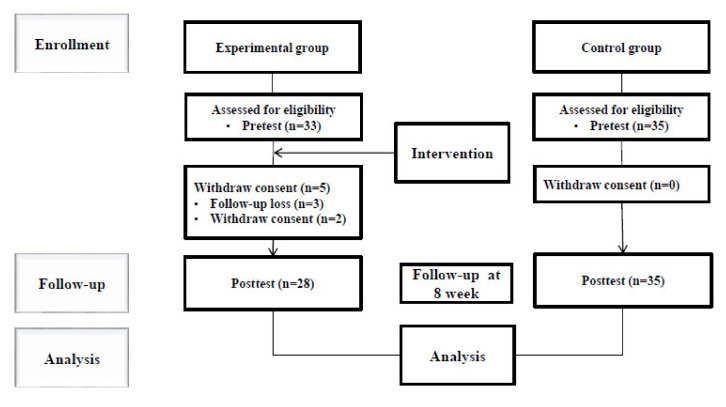
Consort flow diagram.

**Table 1 ijerph-17-06518-t001:** Content of the antiviral agent medication adherence education program (AMAEP).

Session	Methods of Interaction and Transaction	Transaction (Content)	Subject	Time (min)
1 (1 week)	Individual (contact; face-to-face)	Consent form Orientation Pre-survey (questionnaire) Hand out the medication diary, pamphlet	CHC related knowledge	60
2 (2 week)	Individual (non-contact; telephone)	Non-contact education and counseling	Antiviral agents	20
3 (3 week)	Individual (non-contact; telephone)	Non-contact education and counseling	Understanding for medication misuse and bioactive substance	20
4 (4 week)	Group (contact; face-to-face)	Communication among group of 2~5 persons Discussion (group) Check of medication adherence rate (diary, pill count)	CHC and treatment	60
5 (6 week)	Individual (non-contact; telephone) Group (non-contact; SNS ^†^)	Non-contact education and counseling	Management of physical and psychological problems and stress caused by taking antiviral drugs	20
6 (8 week)	Individual (contact; face-to-face) Group (non-contact; SNS ^†^)	Post-survey(questionnaire) Check of medication adherence rate (diary, pill count)	Health care after treatment	60

^†^ SNS = social network service.

**Table 2 ijerph-17-06518-t002:** Homogeneity test for general characteristics and disease-related characteristics of participants in experimental and control Groups (*N* = 63).

Characteristics	Categories	Cont (*n* = 35)	Exp (*n* = 28)	*χ*^2^ or *t*	*p*
		*N* (%)	*N* (%)		
Gender	Male	16 (45.7)	13 (37.1)	0.00	0.578
	Female	19 (54.3)	15 (42.9)		
Age (year)	≤60	11 (31.4)	16 (57.1)	4.20	0.072
	≥61	24 (68.6)	12 (42.9)		
	Mean ± SD	62.91 ± 9.66	60.89 ± 7.67	0.90	0.370
BMI (kg/m^2^)	Mean ± SD	24.02 ± 2.28	23.71 ± 2.31	0.54	0.595
Type of household	Family (spouse and children)	17 (48.6)	9 (32.1)	3.92	0.141
	Spouse	8 (22.9)	13 (46.4)		
	Single	10 (28.6)	6 (21.4)		
Subjected economic level	High	4 (11.4)	4 (14.3)	1.06	0.360 ʃ
	Middle	13 (37.1)	14 (50.0)		
	Low	18 (51.4)	10 (35.7)		
Education level	Elementary school	17 (48.6)	6 (21.4)	7.13	0.028
	Middle & High school	9 (25.7)	16 (57.1)		
	≥college	9 (25.7)	6 (21.4)		
Smoking status	Non-smoker	16 (45.7)	14 (50.0)	0.15	0.458
	Ex-smoker	6 (17.1)	4 (14.3)		
	Current smoker	13 (37.1)	10 (35.7)		
Alcohol	Recommended drinking ҩ	22 (62.9)	18 (64.3)	0.01	0.559
	Heavy drinking ✠	13 (37.1)	10 (35.7)		
Comorbidity	No	8 (22.9)	10 (35.7)	1.26	0.279
	Yes	27 (77.1)	18 (64.3)		
Conmed	None	12 (34.3)	11 (39.3)	0.17	0.794
	Yes	23 (65.7)	17 (60.7)		
AST(IU/L)	Mean ± SD	32.29 ± 16.39	41.68 ± 22.13	−1.94	0.058
ALT(IU/L)	Mean ± SD	38.29 ± 20.96	46.89 ± 25.58	−1.47	0.147
Antivirus drugs	Sofosbuvir + Ribavirin	8 (22.9)	12 (42.9)	3.56	0.168
	Ombitasvir, paritaprevir and ritonavir + Dasabuvir	15 (42.9)	11 (39.3)		
	Sofosbuvir + Daclatasvir	12 (34.3)	5 (17.9)		

Cont = control; Exp = experimental; SD = standard deviation; BMI = body mass index; AST = aspartate aminotransferase; ALT = alanine aminotransferase; ʃ: Fisher’s exact test; ҩ: Male (below 8 cups/week), Female (below 4 cups/week); ✠: Male (above 8 cups/week), Female (above 4 cups/week).

**Table 3 ijerph-17-06518-t003:** Homogeneity test of dependent variables (*N* = 63).

Variables	Cont (*n* = 35)	Range (Min–Max)	Exp (*n* = 28)	Range (Min–Max)	t or Z	*p*
	Mean ± SD		Mean ± SD			
CHC-related knowledge	16.00 ± 4.21	8.00–27.00	15.79 ± 3.59	8.00–23.00	0.21	0.831
Medication self-efficacy	5.83 ± 0.95	4.00–7.85	5.53 ± 0.74	4.00–7.38	−1.17 †	0.245
Medication misuse behavior	6.63 ± 3.44	0.00–13.00	7.25 ± 3.12	3.00–13.00	−0.89 †	0.377
Satisfaction with interaction with healthcare practitioner	20.43 ± 4.38	11.00–28.00	19.93 ± 3.05	14.00–25.00	−0.46 †	0.650

Cont = control; Exp = experimental; SD = standard Deviation; † Mann–Whitney *U* test.

**Table 4 ijerph-17-06518-t004:** Effectiveness of the AMAEP.

Variables	Groups	Pretest	Posttest	F *	*p*	Differences (Post-Pre)	t or Z	*p*
		Mean ± SD	Mean ± SD			Mean ± SD		
CHC related knowledge	Cont	16.00 ± 4.21	16.46 ± 4.48	82.90	<0.001	0.46 ± 1.88	−5.91 †	<0.001
	Exp	15.79 ± 3.59	22.86 ± 3.51			7.07 ± 3.96		
Medication adherence self-efficacy	Cont	5.83 ± 0.95	5.84 ± 0.83	55.08	<0.001	0.01 ± 0.31	−5.02 †	<0.001
	Exp	5.53 ± 0.74	6.90 ± 0.71			1.37 ± 0.92		
Medication misuse behavior	Cont	6.63 ± 3.44	5.86 ± 2.82	53.40	<0.001	−0.77 ± 1.54	−5.00 †	<0.001
	Exp	7.25 ± 3.12	3.89 ± 2.17			−3.36 ± 1.83		
Satisfaction with the healthcare provider	Cont	20.43 ± 4.38	20.06 ± 3.62	109.90	<0.001	−0.37 ± 1.72	−6.61 †	<0.001
	Exp	19.93 ± 3.05	27.60 ± 4.00			7.75 ± 4.57		
Medication adherence rate	Cont		95.66 ± 4.99				−3.88	<0.001
	Exp		99.08 ± 1.32					

Exp = experimental; Cont = control; SD = standard deviation; † Mann–Whitney U test; * F score was from analysis of covariance with pretest scores as covariates.

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
