# Peer review of "Development and Evaluation of an Antiviral Agent Medication Adherence Education Program for Patients with Chronic Hepatitis C"

_ijerph, 2020, doi:10.3390/ijerph17186518_

Round 1
Reviewer 1 Report
General comments: This manuscript is a very important study examining the impact of an education-based intervention to improve medication adherence among patients with Chronic Hepatitis C (CHC). CHC has a significant economic and social burden on patients, and medication adherence for this condition is critical. So, this is an important study. The intervention was grounded in theory and applied well-known principles to engage patients and improve medication adherence. However, I have a few concerns as expressed below. • Drug misuse: The drug misuse construct in the manuscript is not very well explained. In the introduction, drug misuse is merely defined as ‘taking various dietary supplements or over-the-counter medications without consulting a healthcare practitioner’ without any context as to why taking OTC medications is important in the CHC population. I think a little more detail in the introduction can go a long way to make the reader comfortable with why this concept is being measured. In the study methods, I suggest the authors add some information on what questions were asked to assess this behavior. Finally, I would also urge the authors to replace ‘misuse’ with another term. The term misuse is commonly not used for OTC medications, and it often refers to inappropriate use of prescription medications or other substance use. • Methods: The study methods lack clarity on some issues. I have highlighted them below and I recommend the authors add this information into the manuscript o What was the study protocol for the control group? Only the perspective of the treatment group is described in the study. I suggest authors include at least one or two sentences to explain this. o The study protocol explains that a nurse was involved in providing educational sessions with the patient. Was there consistency in intervention delivery, i.e., did the same nurse provide the intervention to all 28 individuals in the study group, or did the researcher include a team of intervention providers? o Why was the final study assessment of medication adherence conducted after 8-weeks even though therapy durations forCHC is often 12 weeks? It would have been beneficial to know medication adherence at the end of the treatment duration. o Were participants provided any incentives to compensate their participation? o The authors mention the content validity index of the AMAEP as 0.88 on page 4, line 131. How was this index estimated? Further details are needed. • Statistical analysis: The analysis conducted for the intervention seem mostly appropriate. However, this was not a randomized control trial and I was surprised to see the authors only conducted bivariate analyses for their primary outcomes. The two study groups were not comparable on variables such as income and education, which can be critical because education is the very focus of the intervention. As sensitivity analyses, the authors should consider • conducting multivariable analyses to rule out the fact that differences in outcomes were not due to differing education levels among the study group. • A brief explanation of the purpose of the PCR qualitative analysis or the Child-Turcotte-Pugh criteria in the methods section could be helpful for the reader. • The term SNS is first used in table 1, but the full-form for SNS is not provided. Please expand abbreviations on first use. This also applies to the abbreviation PCR. • On page 7, line 234, authors use the brand names such as Viekirax and Exvira. Authors should consider adding generic names here in order to make this applicable to an international audience. • In section 3.2 on page 7, please provide the minimum and maximum values for each of the constructs. • I was surprised to see the medication adherence for the control group was high (95%). Reporting previous studies measuring adherence in CHC can help provide context to the 95% estimate and explain why this intervention is important for further research. • I suggest the authors add some sentences in the discussion to address implementation barriers for the AMAEP. Can this intervention be scaled? How much additional resources are expected to required to provide this intervention?

Author Response
I thank you for your thoughtful suggestions and insights, which have enriched the manuscript and produced a more balanced and better account of the research.
Point 1: Drug misuse: The drug misuse construct in the manuscript is not very well explained. In the introduction, drug misuse is merely defined as ‘taking various dietary supplements or over-the-counter medications without consulting a healthcare practitioner’ without any context as to why taking OTC medications is important in the CHC population. I think a little more detail in the introduction can go a long way to make the reader comfortable with why this concept is being measured. In the study methods, I suggest the authors add some information on what questions were asked to assess this behavior. Finally, I would also urge the authors to replace ‘misuse’ with another term. The term misuse is commonly not used for OTC medications, and it often refers to inappropriate use of prescription medications or other substance use.
Response 1: We added to the Introduction section on page 2, lines 45-47. “For patients with HCV, taking various dietary supplements or over-the-counter (OTC) medications constitutes drug misuse and can contribute to liver complications (Nguyen et al., 2008).”
The tool to measure medication misuse behavior was developed by Lee and Park [29]. The questions of this tool are related to the term 'misuse'.
*Add reference; [Nguyen, G. C., Sam, J., & Thuluvath, P. J. (2008). Hepatitis C is a predictor of acute liver injury among hospitalizations for acetaminophen overdose in the United States: a nationwide analysis. Hepatology, 48(4), 1336-1341.]
Point 2: Methods: The study methods lack clarity on some issues. I have highlighted them below and I recommend the authors add this information into the manuscript o What was the study protocol for the control group? Only the perspective of the treatment group is described in the study. I suggest authors include at least one or two sentences to explain this.
Response 2: We added this information to the Introduction section on page 3, lines 95-96.
Point 3: The study protocol explains that a nurse was involved in providing educational sessions with the patient. Was there consistency in intervention delivery, i.e., did the same nurse provide the intervention to all 28 individuals in the study group, or did the researcher include a team of intervention providers?
Response 3: The first author of this study was in constant contact with patients with chronic hepatitis C in the department of gastroenterology and intervention provider. The three assistant nurses performed data collection and control management. The nurses declare no conflicts of interest. They were educated by the first author.
Point 4: Why was the final study assessment of medication adherence conducted after 8-weeks even though therapy durations for CHC is often 12 weeks? It would have been beneficial to know medication adherence at the end of the treatment duration.
Response 4: That was a limitation of this study. The investigator only investigated whether or not they were taking antiviral drugs, which are indicated in the AMAEP. Therefore, we suggest follow-up studies.
Point 5: Were participants provided any incentives to compensate their participation?
Response 5: Researchers met with the participants and provided them with an information sheet explaining the study aims, the confidentiality of personal information, the anonymity of the survey, and the voluntary nature of participation. All subjects (experimental and control group) who completed participation in the program were provided with some incentives (ex, transportation costs, provided tea) in return for participation in the study.
Point 6: The authors mention the content validity index of the AMAEP as 0.88 on page 4, line 131. How was this index estimated? Further details are needed.
Response 6: The content validity indices are as follows:
|
Item |
Expert1 |
Expert2 |
Expert3 |
Expert4 |
Expert5 |
Expert6 |
Expert7 |
Experts in agreement |
Item CVI |
|
1 |
1 |
1 |
1 |
1 |
1 |
0 |
0 |
5 |
0.71 |
|
2 |
1 |
1 |
1 |
1 |
1 |
1 |
1 |
7 |
1.00 |
|
3 |
1 |
1 |
1 |
1 |
0 |
1 |
1 |
6 |
0.85 |
|
4 |
1 |
1 |
1 |
1 |
0 |
1 |
1 |
6 |
0.85 |
|
5 |
1 |
1 |
1 |
1 |
1 |
1 |
1 |
7 |
1.00 |
|
Average I-CVI |
0.88 |
||||||||
Point 7: Statistical analysis: The analysis conducted for the intervention seem mostly appropriate. However, this was not a randomized control trial and I was surprised to see the authors only conducted bivariate analyses for their primary outcomes. The two study groups were not comparable on variables such as income and education, which can be critical because education is the very focus of the intervention. As sensitivity analyses, the authors should consider conducting multivariable analyses to rule out the fact that differences in outcomes were not due to differing education levels among the study group.
Response 7: We added to the Method section on page 6, lines 210-211.
The contents are as follows: “The effects of AMAEP were analyzed using analysis of covariance after controlling for pretest scores.”
We also changed the Results section to include the results of the analysis of covariance on page 7, lines 264-270 and Table 4.
Point 8: A brief explanation of the purpose of the PCR qualitative analysis or the Child-Turcotte-Pugh criteria in the methods section could be helpful for the reader.
Response 8: We added to the Methods section on page 2, lines 83-85.
The contents are as follows: “A widely used criteria to determine the severity of liver injury or presence of fibrosis are the CTP.”
Point 9: The term SNS is first used in table 1, but the full-term for SNS is not provided. Please expand abbreviations on first use. This also applies to the abbreviation PCR.
Response 9: It has been corrected. We changed the text on page 4, in Table 1. We changed the text on page 2, lines 80. The contents are as follows: Social network services (SNS), polymerase chain reaction (PCR)
Point 10: On page 7, line 234, authors use the brand names such as Viekirax and Exvira. Authors should consider adding generic names here in order to make this applicable to an international audience.
Response 10: We changed the medications by generic names on the page 7, line 247 and Table 2.
Point 11: In section 3.2 on page 7, please provide the minimum and maximum values for each of the constructs.
Response 11: We changed Table 3 in the Results section, line 280, to include this information.
Point 12: I was surprised to see the medication adherence for the control group was high (95%). Reporting previous studies measuring adherence in CHC can help provide context to the 95% estimate and explain why this intervention is important for further research.
Response 12: We added more information to the Discussion section on page 10, lines 328-330.
Point 13: I suggest the authors add some sentences in the discussion to address implementation barriers for the AMAEP. Can this intervention be scaled? How much additional resources are expected to required to provide this intervention?
Response 13: We think that in order to address implementation barriers for the AMAEP, future research should focus on well-designed RCTs and high quality studies with more objective assessments of program-related outcomes, as well as more focused investigations of CHC diseases.
Submission Date
26 July 2020
Date of response to comments
13 Aug 2020

Reviewer 2 Report
Title: Development and Evaluation of Antiviral Agent Medication Adherence Education Program for Patients with Chronic Hepatitis C
Summary: I would like to thank the authors for their submission. This paper highlights the development and implementation of a hepatitis C educational intervention to promote medication adherence and improved outcomes with hepatitis C treatment.
Title: Development and Evaluation of an Antiviral Agent Medication Adherence Education Program for Patients with Chronic Hepatitis C
Abstract: Currently written as all one paragraph. If it fits with the journal style, I prefer to have Purpose, Methods, Results, Conclusions separated in order to improve readability.
Materials and Methods: The contents of the program were well described, and I find Table 1 to be helpful. In Table 1, define “SNS”
Results: 3.1 – Gets a bit tedious with all these listed in the text, and also in the table.
Page 7, Line 234: Refer to medications by generic names.
Table 2: Even though medications were not statistically different, are there any differences in the number of tablets per day, side effects, monitoring, etc?
In general, was the genotype of the hepatitis C infections determined, reportable? Were all the medication treatments the same duration?
Page 9, Line 272: 8 cups below/week: Is this the customary unit “cups” should this be “servings” of alcohol? Probably should read Fewer than 8 servings of alcohol per week, Less than 4 servings of alcohol per week, 8 or greater servings of alcohol per week, etc.. Else it is unclear
In general, were the authors able to determine if this increased adherence promoted increases in cure rate/sustained virologic response? Authors should address, as the ultimate goal for these therapies is cure.
Limitations are generally reviewed in the discussion section. Another limitation is time and staff to conduct this intervention. Would this education always be done by a nurse? Would the outcomes be different if education was another discipline, multidisciplinary, etc.
Conclusion: Mention something about potential for cure or desired patient outcomes following this medication intervention. Appropriate to call for additional study.
Author Response
I thank you for your thoughtful suggestions and insights, which have enriched the manuscript and produced a more balanced and better account of the research.
Point 1:
Title: Development and Evaluation of an Antiviral Agent Medication Adherence Education Program for Patients with Chronic Hepatitis C
Response 1: It's corrected.
Point 2:
Abstract: Currently written as all one paragraph. If it fits with the journal style, I prefer to have Purpose, Methods, Results, Conclusions separated in order to improve readability.
Response 2: We were written according to the journal style. The contents are as follows; “The abstract should be a single paragraph and should follow the style of structured abstracts, but without headings”
Point 3:
Materials and Methods: The contents of the program were well described, and I find Table 1 to be helpful. In Table 1, define “SNS”
Response 3: It's corrected. We changed on the page 4 and Table 1. The contents are as follows; SNS (Social Network Services)
Point 4:
Results: 3.1 – Gets a bit tedious with all these listed in the text, and also in the table.
Page 7, Line 234: Refer to medications by generic names.
Response 4: We changed the medications by generic names on the page 7, Line 247-248 and Table 2.
Point 5:
Table 2: Even though medications were not statistically different, are there any differences in the number of tablets per day, side effects, monitoring, etc?
In general, was the genotype of the hepatitis C infections determined, reportable? Were all the medication treatments the same duration?
Response 5: The antiviral agents for the CHC showed a difference dosage of the day, side effects, monitoring etc. by characteristics diagnosed from CHC characteristics. So, we were organized the educational contents differently according to the antiviral agents and made a pamphlet. The duration of antiviral agents use for this study subject was the same as 12 weeks. The selection criteria were as follows; d) patients receiving oral antiviral drug combination therapy by a gastroenterologist
Although not specified in the inclusion criteria of this study, subjects who are planning to take antiviral agents were selected as samples. Thus, the investigator only investigated what they were taking antiviral drugs.
Point 6:
Page 9, Line 272: 8 cups below/week: Is this the customary unit “cups” should this be “servings” of alcohol? Probably should read Fewer than 8 servings of alcohol per week, Less than 4 servings of alcohol per week, 8 or greater servings of alcohol per week, etc.. Else it is unclear
In general, were the authors able to determine if this increased adherence promoted increases in cure rate/sustained virologic response? Authors should address, as the ultimate goal for these therapies is cure.
Response 6: We think, drinking alcohol is a factor that affects the subject's health management and medication adherence. So, we were collected the data by reference criteria (Kim and Shon, 2018). It may be a characteristic variable affecting the rate of medication, which is the ultimate goal of this study.
[Kim, Y. H., Shon, C. H. (2018). Determinants Analysis on Alcohol Consumption Behaviors Focused on Age Effects among Korean Men, Korean Public Health Research, 44(1), 31-47.]
Point 7:
Limitations are generally reviewed in the discussion section. Another limitation is time and staff to conduct this intervention. Would this education always be done by a nurse? Would the outcomes be different if education was another discipline, multidisciplinary, etc.
Response 7: It's corrected. We added the Discussion section on page 12, lines 363-368.
I think a follow-up study is needed to see if another discipline and multidisciplinary can do the program of this study.
Point 8:
Conclusion: Mention something about potential for cure or desired patient outcomes following this medication intervention. Appropriate to call for additional study.
Response 8: We added the Conclusion section on page 11, lines 373-376.
Submission Date
26 July 2020
Date of this review
03 Aug 2020 18:48:52
Date of response to comments
13 Aug 2020

Reviewer 3 Report
The manuscript is a well written article and provides decent platform for further studies. The experiments are properly conducted and explained, and the citations are appropriately referenced. The author(s) acknowledge the limitation in current study and propose future direction for advancement. Moderate English review is recommended.
Author Response
I thank you for your thoughtful suggestions and insights.
Point 1: The manuscript is a well written article and provides decent platform for further studies. The experiments are properly conducted and explained, and the citations are appropriately referenced. The author(s) acknowledge the limitation in current study and propose future direction for advancement. Moderate English review is recommended.
Response 1: We reviewed the contents and English of this study. Thank you.
Submission Date
26 July 2020
Date of this review
07 Aug 2020 12:00:16
Date of response to comments
13 Aug 2020
